# Community-Centered Sustainable Ecotourism Planning in the Bossou Forest Reserve, Guinea, West Africa

**Destina Samani [1], Keith Bosak [2,*] and Sarah J. Halvorson [2]**

[1] West Africa Regional Advisor, United States Forest Service, Accra P.O. Box 16349, Ghana
[2] Department of Society & Conservation, W.A. Franke College of Forestry & Conservation, University of Montana, Missoula, MT 59812, USA
[*] Correspondence: keith.bosak@umontana.edu

**Abstract:** Forest reserves are spaces of conservation and are often spaces of ecotourism as well. Evidence suggests that local community participation in the conservation of the forest reserve space tends to break down under weak ecotourism sector management. The Bossou Forest Reserve (BFR) in Guinea, West Africa has deteriorated considerably due to the fragmentation of the reserve and the inexorable decline in Bossou chimpanzee populations since the 1970s. The situation is largely attributable to several complex and interrelated factors, including the lack of established processes to support meaningful community participation in conservation and ecotourism planning. Ecotourism planning, with the participation of local communities, is considered to strengthen the management of the BFR and conservation connectivity. This paper reports on an approach to sustainable ecotourism planning of the BFR using the Limits of Acceptable Change (LAC) framework to determine the drivers of change and livelihood concerns. Fieldwork undertook in 2019 entailed concept mapping activities which were followed by a participatory geographic information system (PGIS) approach to support community-scale ecotourism planning that considers the multiple needs of stakeholders. Data analysis resulted in a community-centered situational assessment and classification of opportunities, both of which inform ecotourism planning and social-ecological resilience. The study provides baseline data for developing a robust ecotourism management plan capable of coping with the changing internal and external stressors. PGIS-based ecotourism planning can support community priorities and is potentially applicable to other West African areas with similar ecological and livelihood settings.

**Keywords:** Bossou chimpanzees; forest reserve; ecotourism planning; Limits of Acceptable Change (LAC); participatory mapping; Guinea; West Africa

## 1. Introduction

The Bossou Forest Reserve (BFR) located in the West African country of the Republic of Guinea is home to a unique yet critically endangered population of West African chimpanzees (*Pan troglodytes verus*). The Bossou chimpanzees have been the focus of 70 years of primate research as well as the central attraction for the ecotourism [1,2]. Nevertheless, the Bossou chimpanzee population has plummeted since the 1970s from an estimated population of around 25 to a total of 8, thereby placing them on the Red List for Critically Endangered Species [3] This ominous trend is driven by a set of long-term interrelated factors, including habitat loss, forest fragmentation, forest fires and expanding cultivation activities, illegal hunting, indiscriminate poaching, life-threatening chimpanzee diseases, iron ore mining, pressures from refugee encampments, and spatial and biological isolation from chimpanzee populations located elsewhere in the remarkable Nimba Mountains [4,5]. A seemingly expanding zone of the human–primate interface raises fundamental questions about potential actions to reverse the loss of species and to better support conservation goals, ecotourism priorities, and sustainable livelihoods.

The BFR is located in the Guinean portion of the Nimba Range (Figure 1), a transboundary mountain range along the international borders between Guinea, Liberia, and Côte d'Ivoire (7°32′ N to 7°44′ N and 8°20′ W to 8°30′ W) [6]. Several indigenous Manon communities are surrounded by hills 70–150 m high that are covered in primary and secondary forests. Cultivated and abandoned fields as well as secondary, riverine, and scrub forests contribute to a patchy vegetation mosaic. The natural assets of the Nimba Mountains have been of interest to conservationists since colonial times. In 1944 the French colonial government established a strict nature reserve (*forêt classée*)—known as the Mount Nimba Strict Nature Reserve—in Guinea, thereby effectively expanding a protected area that already included the Nimba Range located in Côte d'Ivoire which was protected in 1943. Decades later, this exceptional transboundary area was designated for protection, including as a Biosphere Reserve under UNESCO's Man and the Biosphere Program [7,8]. The designation included a total area of 17,130 ha, with a core protected area of 10,000 ha that extended into the hills of Bossou. This internationally recognized designation unfortunately deprived indigenous communities of their ancestral agricultural lands. In 1981 and 1982, the Mount Nimba Strict Nature Reserve in Guinea and Cote d'Ivoire, respectively, were inscribed on the List of World Heritage Sites. The BFR was not designated as a separate protected area by the Guinean state or international institutions until it was added in 1991 to the "Core Area" of the Biosphere Reserve [9].

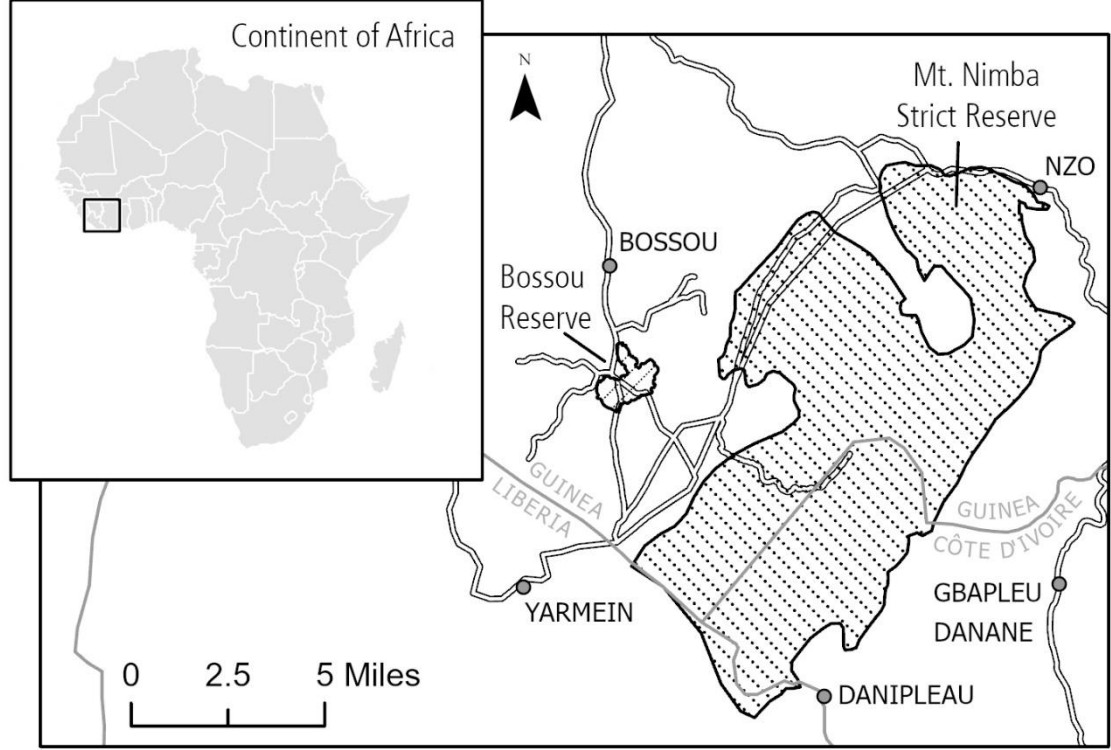

Cartographer: Hannah Shafer, Geospatial Research and Teaching Lab of the University of Montana, 2022

**Figure 1.** Location map of the Bossou Forest Reserve in Guinea, West Africa, located near the transboundary Mount Nimba Strict Nature Reserve.

The BFR is an extremely important area in terms of biological diversity, with many endemic species of flora and fauna. Diverse tree species within the secondary forest flourish, including the umbrella tree (*Musanga cecropioides*), the oil palm tree (*Elaeis guineensis*), and the bush pineapple (*Myrianthus arboreus*). The forests currently sustain relatively few large mammals, as many have fallen victim to past hunting activities. Mammalian species that are hunted today include the red-flanked duiker (*Cephalophus rufilatus*) and the cane rat (*Thrynomys swinderianus*). Forested areas are closed to human activity, some dominated by oil palm trees which are left uncut and serve as critical foraging grounds for

chimpanzees [10]. The BFR (habitat of the Bossou chimpanzees) is in very close proximity to individual livelihoods (such as pineapple, papaya, orange, mandarin, mango, banana farms, palm plantations, etc.) activities in the communities. These community livelihoods are at times consumed/relied on as forage for the Bossuo chimpanzees. This is one major course of human and wildlife conflict in the area.

Re-establishing a flow of migration between the Bossou chimpanzee community and the neighboring Nimba populations will require restoration of the 300 m wide and 4 km long stretch of savanna green corridor created between the BFR and the Nimba Range.

Many residents argue that international conservation groups and government agencies have fallen short of creating opportunities to engage indigenous communities in ecotourism planning and chimpanzee conservation efforts. The lead author has observed mistrust between local communities and the BFR management team during fieldwork. Our observations corroborate written evidence of an obvious lack of local community involvement in ecotourism planning in the BFR. It is now widely recognized in West Africa that local people must be empowered through participatory processes and should be compensated for any losses they incur when protected areas impact their quality of life. Concerns about the imperiled Bossou chimpanzees have led to renewed calls for conservation and protected area management approaches that include local communities in serious planning and decision-making processes, with the long-term goal of social-ecological resilience and the viability of the BFR.

This article reports on a recent effort to foster community engagement in ecotourism planning that involved three communities—Bossou, Nyon, and Serengbara—within the BFR. The study provides evidence for the importance of community-centered ecotourism planning using a Limits of Acceptable Change (LAC) framework and participatory mapping. The research was guided by three primary questions:

- What are the existing issues and concerns of local stakeholders regarding the management of the BFR?
- What are the resources of value within this protected area as perceived by local stakeholders?
- What evidence does the LAC framework provide for sustainable ecotourism planning that would promote a focus on the interests of the community and greater levels of participation in ecotourism planning?

The LAC framework is grounded in social-ecological systems (SES) thinking that considers humans as part of, not apart from, nature [11–13]. In view of the long-standing fractured relationship between the official BFR management and local communities, our assumption is that to fully integrate spatially informed ecotourism planning in the BFR, planners, and managers need to recognize the complexity of the social–ecological interactions in the catchment area and the social processes that influence conservation decisions [14,15]. Recognizing the value of these biocultural spatial relations is crucial if any policy approach to the preservation of biodiversity and chimpanzee protection is to be meaningful to local communities.

## 2. Conceptual Background

The need to build cooperative inclusive relationships with local communities in order to achieve substantive results in the management of forest reserves and other protected areas and biocultural diversity conservation has become widely recognized [16–18]. A conventional approach of demarcating forest reserves with a single borderline has been typified by UNESCO's "Man and the Biosphere Programme", where a strictly protected "core area" is surrounded by a "buffer zone" where sustainable livelihood activities are partially recognized [19]. Under this scheme, which came to be called "community conservation" [15], the inhabitants are presumably encouraged to participate in conservation activities as collaborators. A variety of approaches are used, including employing them as forest reserve employees for their "traditional ecological knowledge", and/or involving them as local representatives in organizations which make decisions regarding reserve management issues [20]. Unfortunately, in many of these "participatory" projects, local

residents participate peripherally in a pre-existing reserve scheme under which administrations and NGOs simply make attempts to enlist local citizens—historically regarded as a "menace"—by presenting them with the carrot of economic profit. Additionally, this mechanism of "participation" still leaves much open space for issues of land dispossession [6].

## 2.1. Chimpanzee Conservation and Society Interactions

Guinea is one of only two countries (the other being Côte d'Ivoire) in West Africa to have known populations of West African chimpanzees; therefore, the country has an essential role to play in the conservation of this species and its transboundary habitat. Zoologist Maxime Lamotte was the first western scientist to identify Bossou as a valuable chimpanzee site in 1942 [5,21]. BFR was established as a long-term chimpanzee conservation biology field site in 1976 by Japanese primatologist Yukimaru Sugiyama from the Kyoto University Primate Research Institute (KUPRI). Since 1976, primate research has been underway, including chimpanzee tool use, cultural behaviors, population dynamics, and life histories [22].

Bossou and the surrounding villages provide a rare example of a site where wild chimpanzees and local people have been living side by side in relative harmony for many generations, sharing the resources of the same forest. Chimpanzees of Bossou are well known for their incredible tool use. They use a stone hammer and anvil to crack open the oil palm nuts. In addition, they use various tools to reach and access foods that would normally be inaccessible, to defend themselves, to communicate, to explore their environment, to maintain hygiene, and to engage in ant-dipping in ways that appear to be unique to this community. They have an exclusive range of behaviors that include pestle-pounding, algae scooping, and dismantling traps. Researchers suggest that their knowledge of how to detect and dismantle these traps is passed on to younger members of the community [10,23–25].

The number of Bossou chimpanzees has decreased to seven adults (three males and four females) and one juvenile in recent times. The Bossou chimpanzee community remains isolated from neighboring groups of chimpanzees in the Nimba Range; no females have transferred to the BFR from neighboring communities in four decades. Three out of the four females in the Bossou community are estimated to be over 50 years old, which is the life expectancy of chimpanzees in the wild [26]. Their genetic diversity and indeed their survival are now critically threatened. The isolation of their habitat and the associated insufficiency of genetic interaction and gene flow between neighboring populations presents serious conservation problems [24].

Indigenous communities have long-standing forest, cultural, and spiritual connections to the Bossou chimpanzees, with a rich oral history that details relatively harmonious human-chimpanzee interactions and traditions over decades. The Manon people in several farming communities (population 2500) have been living in relative harmony for many generations with chimpanzees, sharing the resources of the same forest. The chimpanzee is a totem animal of the most influential family; hence, hunting or eating the chimpanzee is strictly forbidden. Local people consider the chimpanzees as the reincarnation of their ancestors [27].

Manon subsistence farmers continue to practice intensive swidden techniques on individually owned land or leased land, to grow rice, cassava, and a wide variety of fruits such as pineapple, papaya, orange, mandarin, mango, and avocado for local consumption and commercial markets. Increasingly, there are several meeting nodes between the chimpanzees and humans—both friendly and confrontational—across several aspects of everyday life. One overlap is the palm oil trees, given that both populations rely on palm fruits. Local people and chimpanzees rely heavily on oil palm nuts, yet competition is low because of the high density of the oil palm as a communal resource. The chimpanzees regularly crop-raid and are known to attack villagers occasionally. People's reactions to crop raiding vary in the village; some tolerate it as they believe the chimpanzees have a right to feed off a small number of their crops, while others actively chase and throw stones at any chimpanzees seen near their farms. Local communities observe that the presence of

papaya trees brings chimpanzees into close proximity to people's houses, and this is perceived to increase the likelihood of chimpanzee attacks on humans. Some people have cut down papaya trees located near the forest edge in an attempt to reduce human-chimpanzee contact. Pineapple raiding is rarely tolerated, and humans heavily guard these cultivated areas with the use of dogs. The chimpanzees are opportunistic frugivores and their ability to adapt to new situations has allowed them to exploit agricultural land that impinges upon their natural habitat. Although the Bossou chimpanzees are afforded some protection and tolerance due to the cultural beliefs of the local Manon people, their future is uncertain as their habitat becomes fragmented and threatened [24,27].

Community members and chimpanzees spatially interface through their respective mobility. Bossou Village and the chimpanzees' home range are dissected by one large road (approximately 12 m wide) that stretches from the Guinea-Liberia border into the forested region of Guinea. This road was expanded in December 2004 and is frequently used by large trucks, cars, motorbikes, and pedestrians. A narrower dirt road (approximately 3 m wide) branches off from the large road; the smaller road is used by pedestrians. The chimpanzees must cross both roads to move from one forested area to the next. Both roads have forest cover up to the edges and are separated by a middle zone of secondary forest and coffee plantations. The chimpanzees typically cross the two roads at specific points that the group has used for decades [10]. However, road-related accidents are not a major source of mortality for Bossou chimpanzees.

## 2.2. BFR Management and Governance Challenges

Currently, the BFR is managed through a state-private sector collaborative agreement involving the Bossou Environmental Research Institute (Institut de Recherche Environnementale de Bossou, IREB), Directorate General of Scientific Research, Innovation and Technology (Direction Nationale de la Recherche Scientifique et de l'Innovation Technologique, DNRSIT), and KUPRI. Since 1995, the Nimba Mountains Environmental Management Center (Center de Gestion de l'Environnement des Monts Nimba, CEGEN) has also contributed to oversight of the Nimba Biosphere Reserve. CEGEN is a public organization of administrative and scientific nature, with a mission to coordinate and promote activities aimed at protecting the World Heritage Site and the rational use of the biological resources of the Nimba Range.

In October 2001, IREB was established under the Ministry of Higher Education and Research to promote environmental conservation and scientific research. The establishment of IREB and subsequent appropriation of indigenous lands for conservation purposes contributed to open resistance and conflict between the local people and the organization. In addition, there were disagreements over the unsatisfactory swidden farming practices of the local people, which, among other concerns, contributed to forest fires in the conservation area. Some employment opportunities were realized by local people who have been employed as nature guides in the conservation area.

Prior to IREB's involvement, there was no governmental organization in Bossou in charge of environmental conservation and tourism. The plan for IREB was, in addition to receiving foreign researchers and conducting research with the Institute's own researchers, to place the various interests related to chimpanzees (such as guide employment, consolidation of infrastructure through individual aid, and allocation of tourism income) under the Institute's control. As such, among other initiatives, IREB organizes paid-for chimpanzee tracking expeditions. According to IREB, it allocates half of the revenue accrued from visitor fees to the community. IREB prescribes the rule of engagement and conducts for visitors to the Reserve.

Recent interventions have focused on reforesting the deforested region that cuts off BFR from the larger Mount Nimba Reserve. The KUPRI-International researchers, in collaboration with IREB, the villagers, and local NGOs, initiated a reforestation program called the Green Corridor Project in 1997. This project has the aim of planting trees to enlarge the forests of Bossou and create a corridor over the savanna that separates the hills

of Bossou from the Nimba Mountains. Since 1997, researchers from KUPRI have been working in cooperation with IREB and local villagers, and with the support of the Japanese Embassy in Guinea, the Guinean government, and various international organizations, toward the creation of a "green passage" [28]. The aim is to plant trees along a 300 m wide and 4 km long stretch of savanna extending between Bossou and the Nimba Mountains in the hope of re-establishing a flow of migration between the Bossou chimpanzee community and the neighboring Nimba populations. The participating villages are Bossou, Serengbara, and Nyon.

Increasing the connectivity between fragmented forest blocks is an innovative approach to conserving genetic diversity and ensuring the survival of chimpanzees, as well as other animal and plant species. The corridor project is planting saplings, at a density of 400 young trees per hectare. Priority is usually given to savanna-adapted species such as sugar plum (*Uapaca guineensis*), African Locust Bean (*Parkia bicolor*), and Guinea Plum (*Parinari excelsa*). A new technique involving hexagonal plastic tubes, or "hexatubes", has been used since 2005 to protect young trees against grassland invasion, herbivores, and dehydration.

There is a clear and obvious lack of community interest in the Green Corridor Project. Interactions with IREB, corroborated by the three affected villages, indicate that local people feel marginalized to the extent that they demand financial compensation for attending meetings convened by IREB and the Green Corridor Project team. This is believed to stem from the apparent alienation of the village headmen and council of elders in the planning process. As such, they largely view themselves as invited guests and not partners.

### 2.3. Limits of Acceptable Change Framework

The Limits of Acceptable Change (LAC) planning approach [29] was initially designed to address visitor management issues in protected areas in the United States and was a product of the growing realization that the concept of carrying capacity failed in achieving its objectives [30]. While there are many reasons why the carrying capacity paradigm failed, the most fundamental was that it impelled managers toward the wrong question: "How many is too many?" Carrying capacity is intrinsically a quantitative term; however, research has shown that many problems of recreational use are more a function of the behavior of people rather than sheer numbers of people [30]. The LAC framework, on the other hand, deals with a significantly different question: "What resource and social conditions are appropriate (or acceptable), and how do we attain those conditions?" This question represents a substantially different approach to thinking about tourism and recreational use.

The LAC is a framework for protected area planning within which decisions can be made about the kinds of conditions that will be permitted to occur in an area. The basic premise of the LAC framework is that change is a natural, inevitable consequence of recreational use. Both environmental and social changes are involved. At the heart of this is the question: How much change is acceptable? The LAC framework recognizes the inevitable impacts that occur from human use. It seeks to address the conflicting positions between managers and the user behavior of patrons and local communities [29]. Although the LAC framework was originally developed for use in the U.S. Wilderness Reservation System, it has been used widely with success in many areas of the world [31–34] (Boentoro et al., 2021, Diedrich et al., 2011, Dragovich and Bajpai 2022, Vann 2005). LAC offers a context for a broader spectrum of public participation in nature-based regional planning [30]. Public participation is important for sustainable planning, given the interconnected nature of community livelihoods and the BFR, especially with the easement provided for local use of ecosystem services such as land and water to support farming [35].

LAC can be updated as more information becomes available to ensure they accurately reflect the natural variability (or normal range for artificial sites) around the time of listing critical components, processes, and benefits or services of the conservation area. If a site has improved or has been restored and is being managed to maintain an improved baseline, it may be appropriate to review and update the Limits of Acceptable Change to ensure they reflect the new baseline [29].

The LAC framework is rooted in the premise that change is a natural, inevitable consequence of human-induced factors such as recreational use and farming practices, among others. As such, the question should be how much change an area can tolerate and what strategies should be taken to avoid unacceptable impact, in view of the pre-established tolerable impact thresholds for the conservation area [29].

Our application of the LAC framework was restricted to the first two steps: (1) Identify area concerns and issues (situational analysis); and (2) Define and describe opportunity categorization (resource values assessment) through the process of identifying the existing opportunities in the area, then categorizing them.

Locally grounded data was sought that would reflect the issues and values expressed by the stakeholders (managers, scientists, the general public, and local communities) in a process aimed at developing the foundation for a community-centered ecotourism plan.

LAC Framework—Situational Analysis

1.   Are there notable conservation, cultural, religious, scientific, historic, or educational values in the area that demand special consideration?
2.   What is the collective decision of the three communities regarding the issues that demand special consideration?
3.   What is the state of biodiversity in the area: threatened, endangered, or secure?
4.   What are the connections (social, economic, and cultural) among the three participating communities and other adjoining communities?
5.   Are there unapproved uses/practices that demand special consideration?
6.   What are the current concerns of the surrounding/indigenous communities about the management of the reserves?
7.   What are the community expectations of the kind of management they want to see?

LAC Framework—Resource Values Assessment

1.   What natural resources are of value to the community?
2.   Where are the listed natural resources located in the community?
3.   Which of these resources are within the areas of the chimpanzee reserve?
4.   Which of the valued natural resources listed are threatened?
5.   Which of the valued resources have the potential for tourism for development?
6.   How can a community-centered approach support the establishment of resilient processes and structures in the management of the reserve?

It is widely recognized that documenting resource values has proven to be a compelling way to help elevate ecological and cultural priorities in decision making and management forums. Our approach builds upon a large body of work that demonstrates the importance of resource values in guiding ecotourism development that aligns with conservation outcomes [36,37].

## 3. Materials and Methods

The BFR is a complex social-ecological system that exemplifies both environmental and sustainability problems, characterized by scientific uncertainties, multiple interrelationships between human and non-human actors, non-linear dynamics, large-scale consequences, and irreversible damage. These problems call for a highly participatory approach with stakeholders, particularly local communities, to ensure sustainable ecotourism planning [38–40]. As such, our broad aim was to undertake a situational analysis of the BFR, with attention to resource values as perceived by residents of the three primary communities adjacent to the BFR. We created participatory components to correspond with elements of the LAC framework guiding our approach. These participatory approaches—concept mapping and participatory geographic information system—followed established protocols [41] and were informed by successful applications in other conservation and ecotourism planning contexts [42–45]. As part of the open call for public participation in community forums, specific steps were made to reach certain individuals and groups who are often underrepresented and could provide unique perspectives for consideration. These

included women and community leaders. Participants were not incentivized. We carried out this work by building on the long-standing partnership between the communities, IREB, AUDER, the USFS, and our research team members. The communities also saw the intervention as contributing to improving the ongoing ecotourism in their area.

AUDER, the local implementing partner of the U.S. Forest Service in Guinea, facilitated the recruitment of local participants for the workshops. The field team deliberated and resolved specifics of the fieldwork: selection of local translators (French and Manon speakers) and facilitators; the design and execution of the training exercise; and the simulation and scenario modeling of community workshops using local materials anticipated to be accessible in the actual community workshops. As part of the preparatory process, there were preliminary community engagements in all three participating communities.

### 3.1. Concept Mapping

The concept mapping workshops were open to all members of the three communities that interact with the chimpanzee corridor. Across all communities, the workshops attracted workers in the plantations, traditional healers, hunters, herders, local forest guides, merchants, and others. Participants were selected for their diversity in livelihood strategies. The sample is not representative but rather designed to elicit a diversity of perspectives across communities. Approximately one-third of the participants were women. Each workshop lasted about 3 h. In the concept mapping process, each community's output map of natural resources of value identified was used as foundation material to classify the opportunities. Participants were asked to mark out or vote for (using writing markers) a list of valuable resources to communities. The votes were tallied, ranked, and three top-priority resources were selected for the concept mapping exercise (Table 1). The participants in each community workshop were organized into breakout groups of three (males and a female). In each case, the groups were guided to select one of the three valued resources, draw out and discuss the perceived and real disturbances and how those disturbances impact the selected resource (Figure 2). Each group was given a large sheet of paper, colored markers, and post chits to create collective causal loop diagrams. Facilitators helped participants develop causal loops based on the DPSIR framework, which identifies the Driver-Pressure-State-Impact-Response. Causal loop diagrams were created to define the linkages, identify the drivers of change, and highlight the effects and impacts on the resource. A color-coding scheme was adopted to help illustrate and highlight the drivers of change and associated perceived consequences. Facilitators guided participants to compare and discuss all three group concept maps with the goal of identifying the commonalities, the different interconnected variables, and the ways change in one place could affect other resources.

**Table 1.** Highlights of community-specific workshops and priority resources identified.

| Community | Representation Summary | Priority Resource of Value |
|---|---|---|
| Bossou | 19 participants Comprised hunters, traditional healers, forest guides, farmers (cultivators), people who work within the corridor (market women among others) | Priority 1: Hills Priority 2: Plantation Priority 3: Corridor Others: Sacred forests, rivers |
| Nyon | 18 participants Comprised hunters, traditional healers, forest guides, plantation farmers (cultivators), people who work within the corridor (market women among others) | Priority 1: Plantation Priority 2: Forest Priority 2: Corridor Others: Rivers, bat caves, sacred forest, water spots, savanna, natural bridge |
| Serengbara | 18 participants Comprised hunters, traditional healers, forest guides, plantation farmers (cultivators), people who work within the corridor (market women among others) | Priority 1: Plantation Priority 2: Forest Priority 3: Corridor Others: Rivers, sacred forest, bat cave, natural bridge, tree nursery (corridor), savanna |

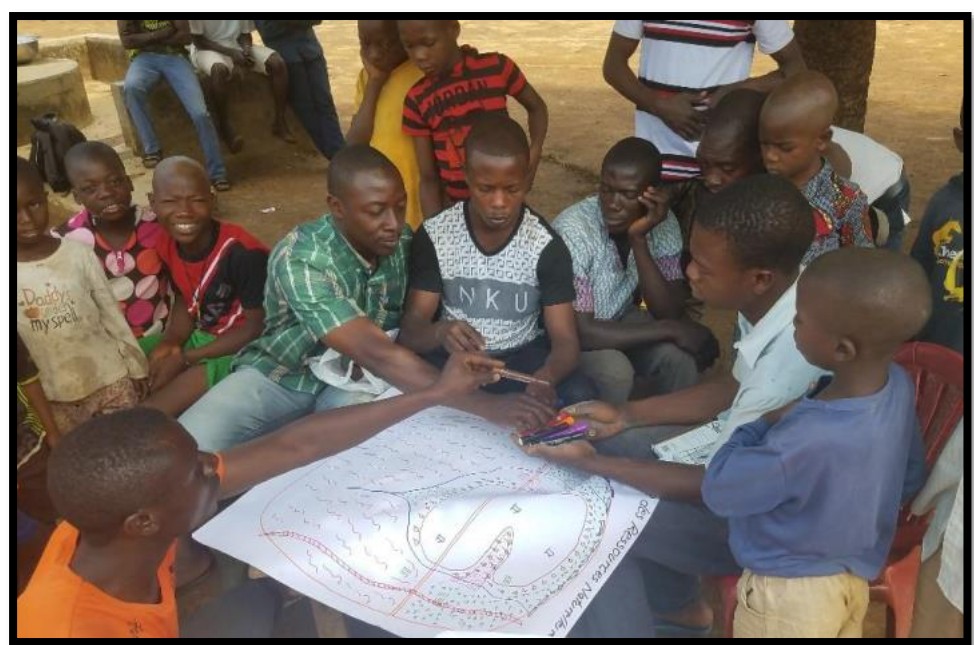

**Figure 2.** Concept mapping session with youth in one study community. Participants are transferring the data from the on-the-ground dirt map onto paper.

The second data collection method built upon the causal loop diagrams developed in the small groups and centered on concept mapping in which participating communities (Bossou, Serengbara, and Nyon) were taken through a series of brainstorming sessions to draw out the underlying concerns and the drivers of what they perceive of as existing challenges in the BFR (Figure 3). The community activities enabled participants to reach a consensus largely through a rigorous constructive debate on the important values associated with the BFR. Facilitators (three from the research team and three local facilitators) were solidly prepared for engaging participating community members having analyzed data and information shared by field guards and leadership team members of IREB and secondary data in the form of technical reports, scientific studies, and historical and policy documents.

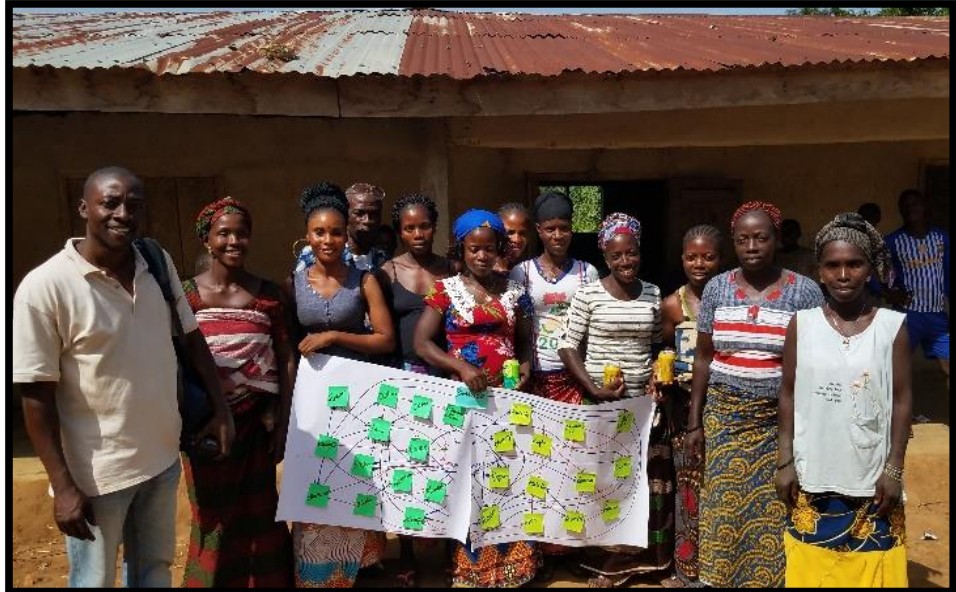

**Figure 3.** Women-specific concept mapping in breakout group with facilitators.

The concept mapping drew up the relationship between the different variables, including those that strengthen or weaken the argument for the restoration of the Corridor and the development of the ecotourism potential of the Bossou Forest Reserve and its contiguous areas.

In addition to the community validation workshops, the project team engaged identified stakeholder groups and entities throughout the country such as the Ministry of Environment, the Ministry of Tourism, Hotels and Handicrafts, and the Ministry of Higher Education and Scientific Research; IREB, CEGEN, and AUDER. The interactions provided the opportunity to share the preliminary findings of the team and gather valuable feedback to aid the project.

### 3.2. Participatory GIS

Participatory system mapping provides a community-centered pathway to assess the complexity of the system [43]. The utilization of the Participatory Geographic Information System (PGIS) for community resource mapping engenders transparency, greater participation, inclusion, empowerment, and ownership of the local spatial information, and it also gives the public a meaningful stake in the decision-making processes. Specifically, the PGIS equips the participating communities to solve spatial planning problems on their own with limited facilitation by the project team. We sought to ensure the representation of resource values and to support local efforts to incorporate those values into the conservation and ecotourism planning and decision forums.

Mapping workshops were organized in the three participating communities on separate days. Workshops drew a variety of community groups including teachers, herders, hunters, opinion leaders, youth groups, religious leaders, and women; altogether, 144 individuals participated (Table 2). The mapping workshops included a larger portion of the local population than the concept mapping exercises and built upon the results from the concept maps.

**Table 2.** Participation in mapping workshops by community members.

| Community | Estimated Duration | Attendance |
|---|---|---|
| Serengbara | 5 h (morning) | 54 men, women, and youth; estimated 30% women |
| Nyon | 4 h (afternoon) | 48 men, women, and youth; estimated 25% women |
| Bossou | 4 h (morning) | 43 men, women, and youth; estimated 20% women |

Step 1: Identification of Resources of Value

Individual participants were encouraged to list all the natural resources that are considered to have spiritual, commercial, and/or touristic value. Each of the identified resources considered collectively to be of value was documented on a chit of paper and handed over to the individual who made the list as evidence of their participation. In view of the heavy patriarchal culture in this area, women and youth were tactfully prioritized by the facilitators to capture their marginalized voices. The identified resources formed a foundation for the next stage of the workshop, as these resources were aggregated to create a listing of collectively held values.

Step 2: Production of Participatory Maps

Participants were organized into community-specific sessions and invited to discuss and portray the state of their communities' natural environment which forms part of the Bossou Forest Reserve. In all cases, the facilitator worked cooperatively with each community to draw a baseline community boundary in the bare dirt on the ground. One after the other, participants submitted chits of paper (containing identified valued resources) to the facilitator for consensus on where to locate on the on-the-ground map the particularly

valued resources included in the list. Locally available materials such as sawdust, tomato powder, blue dye, white powder chalk, plants and seeds, stones, rocks, river sand, and other items were used to represent specific resources on the map. A key component was identifying the resources of value to the entire community. The workshop confirmed areas and features of value (e.g., sacred areas, developed and latent touristic features) as well as ecosystem services and threatened areas of existing or potential significance. Using the actual natural landscape, each feature's position was collectively determined in reference to specific baseline features in the physical environment.

A map was formed by positioning and fixing the established features on the ground (Figure 4). In all communities, but particularly in Serengbara, the process was characterized by animated discussion, marked by open disputes, contestations, and at times verbal confrontations, mostly relating to the disputed boundary of the BFR. After positioning all the listed valuable resources on the map, the next step was to highlight the resources of value that were threatened based on historical accounts and personal observation. Consequently, participants through discussion and consensus building agreed on resources with the potential for ecotourism objectives within their respective communities.

Step 3: Recap and Knowledge Sharing

At the end of the ground mapping exercise, a participant from each community (who demonstrated a fair understanding and involvement) was encouraged to volunteer to explain the output map in the Manon language to all participants. These exercises were also used to gauge the extent of community participation and the understanding of the process and the map. In all the communities, three lead participants, with a fair understanding of the mapping process were nominated to join a designated project team member in transferring the details of the on-the-ground dirt map onto a large-sized paper (flip chart), later to be superimposed on a pre-developed community base map. Again, this stage of the process ensured an unbroken chain of community involvement in the process.

Step 4. Development and Validation of ArcMaps

The on-the-ground dirt base maps developed by the communities were given to a GIS technician to be transformed into digital ArcMaps for the respective communities. A technical validation workshop was then held with each community nearly two months after the community mapping exercise. Each validation workshop included selected community leaders native to the area and who participated in the initial community mapping exercise. The main objective was to evaluate and technically validate the digital maps produced from the on-the-ground dirt base maps. The workshop format was guided by open-ended questions and an iterative process. This step was driven by a spirit of consensus, resulting in agreement about the map revisions and the final versions that were then reviewed by the communities.

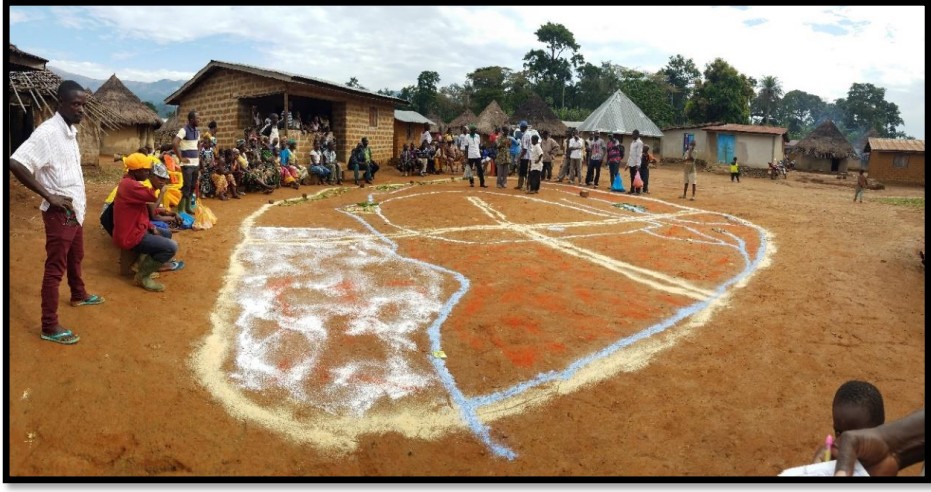

**Figure 4.** Participatory mapping using local materials in Nyon.

The proposed maps were presented to the respective communities on large format banners (1 m × 1 m). They were large enough to allow for clear viewing and editing by the participants. Participants were taken through the maps, and the ensuing open debates were animated and covered a broad range of issues and areas needing modification. One area of serious contention was the delineation of the BFR boundary in relation to the communities, particularly in the case of Serengbara. Recommendations for modifications to the maps were collectively accepted for review and the maps were adopted by participants as a working document to guide future work for community conservation activities and future assessments.

The ArcMaps were edited based on feedback from the community validation workshops, and subsequent changes were carried out using ArcGIS. A final map representative of the collective thoughts of the participating communities drew particular attention to the chimpanzee corridor (Figure 5).

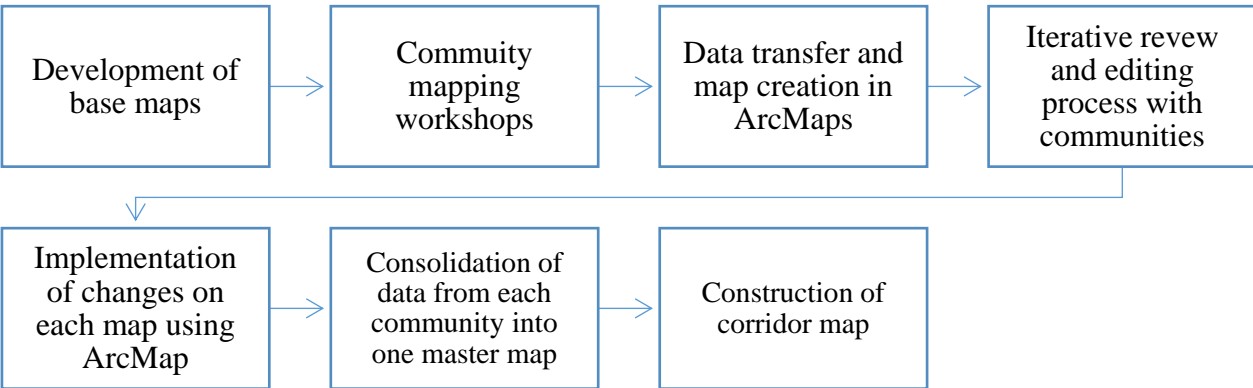

**Figure 5.** Workflow of the participatory GIS during spring 2019.

From these community encounters, residents identified areas within the BFR that are considered to be off-limits to visitors due to cultural and spiritual significance. Significantly, the community engagements in concept mapping and PGIS were based on transparency, trust, and a commitment to fostering open and inclusive social space to empower the indigenous people to participate in the shared solution. Lead individuals from each community were selected to participate in the process of transferring and transforming the hands-on representation into three specific community base maps (See Figures 6–9). The PGIS facilitated and augmented data collection and allowed a recursive and reflective approach to be taken to both the output maps and the resulting conceptualizations of biocultural spaces and ecotourism planning opportunities [46].

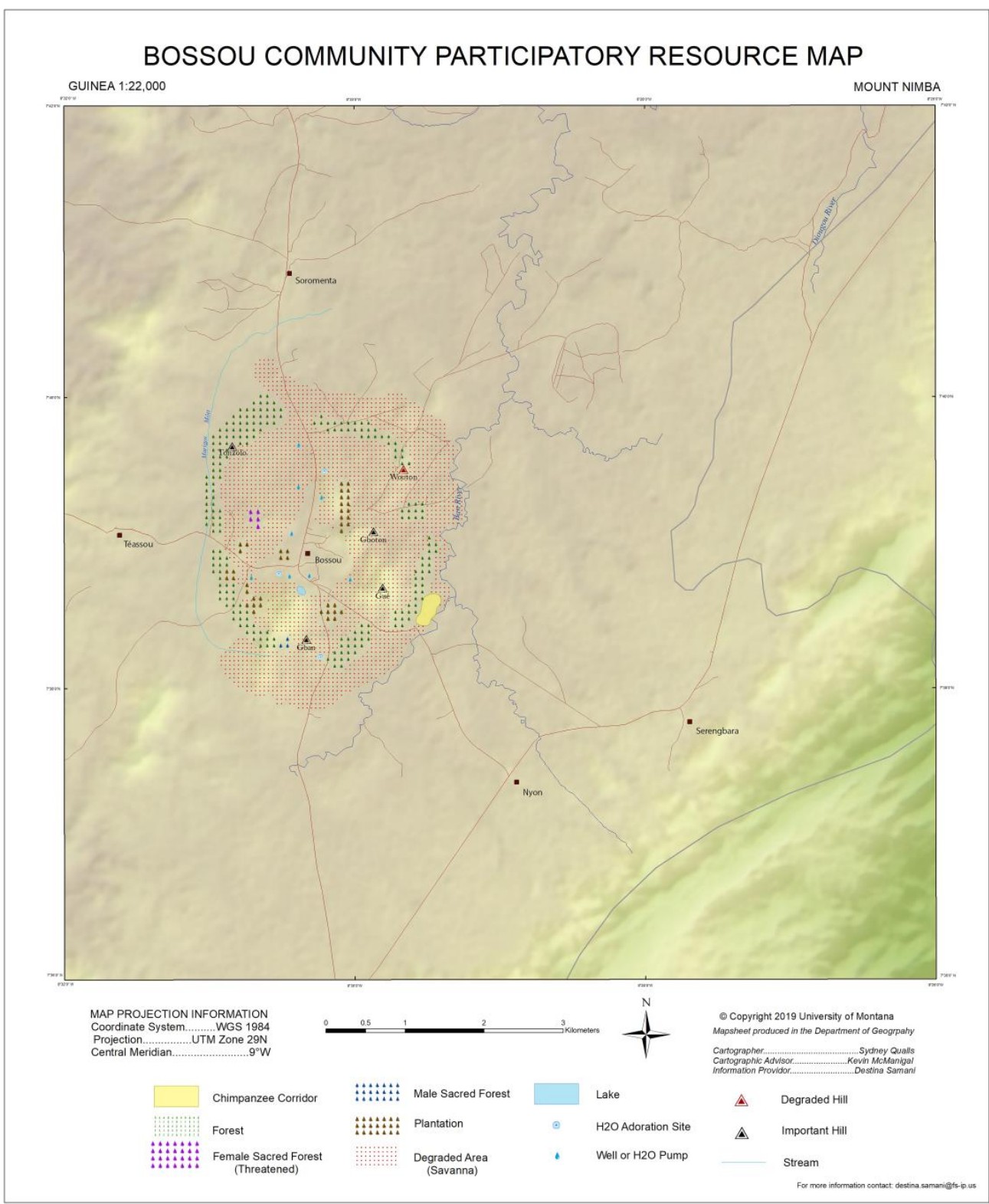

**Figure 6.** Bossou Community Participatory Resource Map.

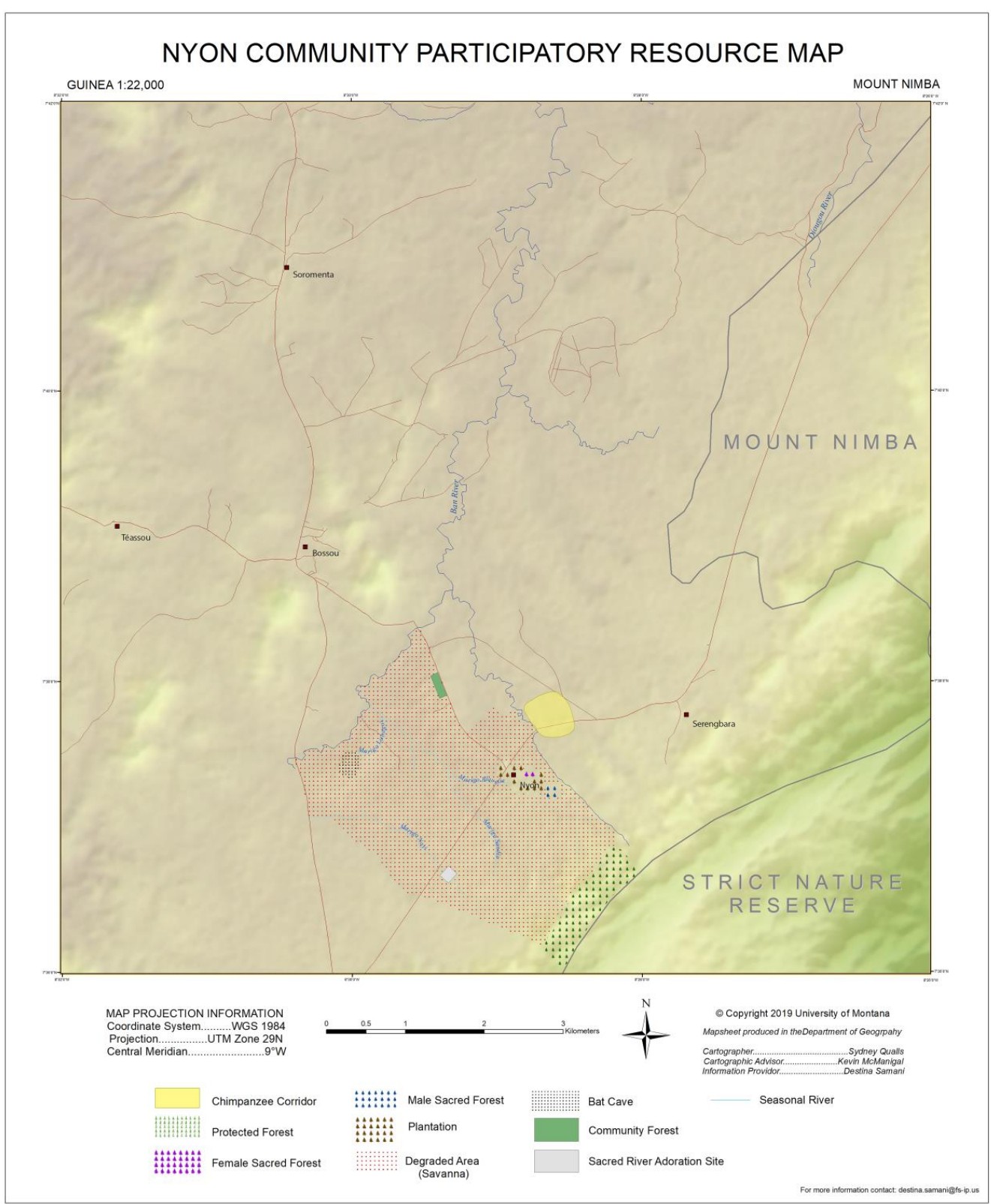

**Figure 7.** Nyon Community Participatory Resource Map.

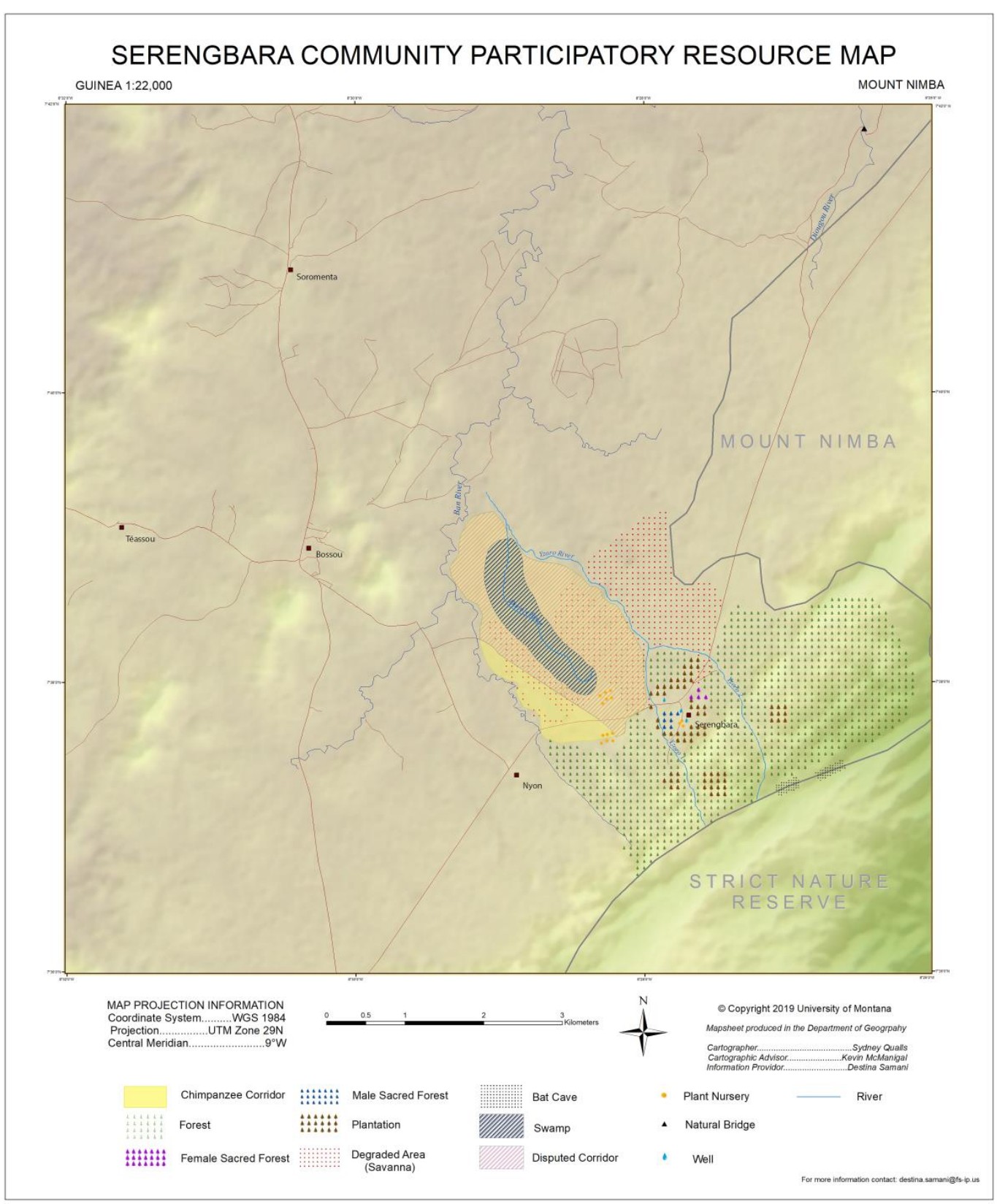

**Figure 8.** Serengbara Community Participatory Resource Map.

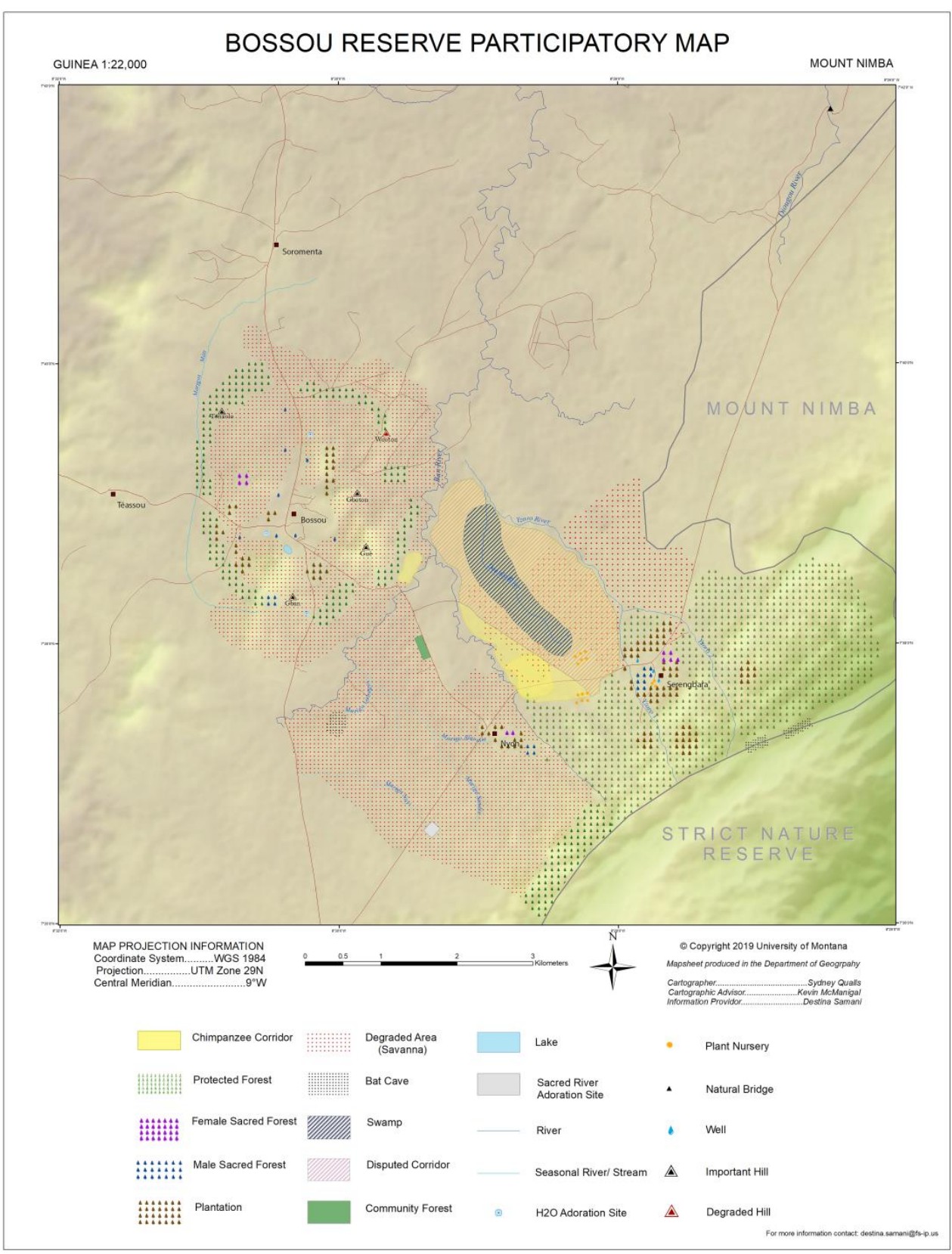

**Figure 9.** Bossou Forest Reserve Participatory Map.

## 4. Results

### 4.1. Outcomes from Concept Mapping

The concept mapping workshops captured community members' representations of complex drivers of change as well as cause-effect relationships affecting natural resources (Table 3). Results were structured using the Driver-Pressure-State-Impact-Response (DPSIR) framework, which helps to describe the interactions between society and the environment [47].

**Table 3.** Summary of results of concept mapping in three study communities.

| Driving Forces | Pressures Induced by Human Activities | Impacts of Environmental Degradation | Responses at Institutional and Local Scales |
|---|---|---|---|
| Agriculture<br>Charcoal production<br>Herding<br>Immigration<br>Conflict<br>Tourism<br>Mining<br>Culture<br>Tradition | Climate change<br>Famine<br>Air pollution<br>Wildland fires<br>Land erosion<br>Over-exploitation of ecosystem services | Illegal tree logging<br>Loss of biodiversity and habitat loss<br>Farm raids by chimpanzees<br>Hunting of chimpanzees<br>Disease infection by chimpanzees<br>Loss of forest cover between Bossou and wider Nimba Reserve<br>Lack of gene flow between Bossou chimpanzees and wider Nimba area | KUPRI-International Team Green Corridor Project<br>USFS Conservation Intervention Programme<br>Local community fire brigades |

The BFR is facing major threats to the conservation of its biodiversity. These are mainly human-induced factors, either directly emanating within or outside the boundaries of the protected area. The increasing development of Bossou and Nimba areas, through population growth and the opening up of communities for commercial activities, has expanded the non-human and human primates' interface. This interface is characterized by chimpanzee crop raiding owing to the loss of natural habitat. There is testimony from local people that, sometimes, people, particularly non-natives, feel threatened by wildlife due to fears about crop loss and also personal safety [48]. Consequently, in an attempt to curtail such commercially threatening activities, farmers may attack (shoot) primates feeding off agricultural land in order to protect their crops, irrespective of their protected status.

In Guinea, habitat destruction resulting from agricultural development and commercial logging has been identified as one of the most important factors affecting biodiversity and chimpanzee survival, by leading to the fragmentation of forest and therefore isolation of the populations. Two main factors have been identified as responsible for environmental degradation. The main agricultural method used in Guinea is the slash-and-burn culture associated with mono-cropping. Slash-and-burn agricultural exploitation has a strong impact on the environment: parts of the forest are yearly cleared and burned, leaving vast surfaces of fallowed land. Core areas of the Mount Nimba Strict Nature Reserve are not spared in this process, which is resulting in a loss of general biodiversity [49,50]. Contextual reasons such as demographic growth, farming tools, modernization, and a critical economic situation are contributing to accelerating this phenomenon.

Illegal hunting and poaching occur within the forests of Bossou; gunshots are often heard and traps are frequently confiscated by guides. Several motivations can lead a local person to hunt a chimpanzee. Their meat is appreciated by many people in Guinée Forestière, although specific ethnic groups such as the Manon people have cultural and traditional taboos against eating or killing this animal. Some cultivators report chimpanzees destroying their crops. In some cases, the chimpanzees are killed for that reason. In a period of fruit scarcity (at the end of the dry season), it is difficult for chimpanzees to find enough food resources in the forest. Reduction and fragmentation of the natural habitat are also making this situation worse, pushing chimpanzees towards the plantations and fields.

Many places across the reserve are under pressure from potential mining of iron ore, habitat destruction mainly due to cultivation, and indiscriminate poaching using snares and firearms. The purest iron ore deposit is localized in the northern part of the Nimba Mountains (Guinean part) where a mining enclave has been delimited. Mining would have

dramatic consequences on the environment and on chimpanzees in the region. The resulting influx of people will increase the pressure for cultivation, causing a reduction in fallow time, further deforestation, and higher habitat encroachment. In addition, mining would likely cause erosion, water pollution, and consequent deterioration of swamp-agriculture rice yields. This would severely affect the local population, which would search for other income and possibly increase poaching, hunting, and felling of the forest.

The main cause of death in chimpanzees at many long-term research sites is an infectious disease. Because chimpanzees and humans are so similar, chimpanzees may contract many pathogens that are either simply carried by humans or that afflict humans (e.g., tuberculosis, polio, pneumonia, typhoid, and Ebola hemorrhagic fever) [51]. If not properly managed, conservation research and tourism activities present an elevated risk of disease transmission for both species. Chimpanzees are extremely vulnerable to respiratory diseases which have become a major cause of death, especially among habituated populations. In November 2003, all members of the Bossou community suffered from an outbreak of a respiratory disease which resulted in the confirmed death of four individuals (two infants, one adolescent male, and one old adult female) and the presumed death of one old adult female. The only other confirmed respiratory disease outbreak at Bossou since 1976 occurred in 1992 and resulted in the death of an infant. The 2003 epidemic strongly reminds us of the vulnerability of chimpanzees to human-borne diseases, especially respiratory diseases, and the urgent need to put in place practical measures aimed at preventing the occurrence of similar outbreaks in the future (for example, the compulsory wearing of masks when within 20 m of the chimpanzees).

### 4.2. Outcomes from PGIS Workshops and Related Activities

PGIS contributed original knowledge and creative insights into how local community members conceptualize and resiliently inhabit this landscape in relation to their own cultural models, community priorities, and connections to species of concern such as the chimpanzees.

Notable conservation, cultural, religious, scientific, historic, or educational values demand special consideration:

The Bossou and Nimba Biosphere Area hold significant tourism potential. The BFR is noted for its chimpanzees. It is a draw for academics and students (and an international team) conducting scientific research into chimpanzee behaviors among others. Ecotourism is quite integrated into the economic structure of the local communities; many have livelihoods tied to this sector as tour guides, translators, and conservation field officers. Moreover, visitor inflows, mobility, and long stays in the local communities contribute to visible knock-on effects such as the proposed expansion of existing accommodation services for tourists and the sale of food and cultural artefacts.

The adjoining area comprises bat caves, rivers, natural bridges, and impressive landforms that have the potential for development. The plantations are a vital source of sustenance for the communities and a sustainable strategy for conserving the chimpanzee corridor against wildfires. It provides their food and nutritional needs and presents an opportunity to safeguard further degradation of the protected areas from annual farming practices. All communities have a spiritual attachment to the forest and some have cultural connections to local rivers that inspire interest in, and actions that ensure, river protection.

Availability of critically threatened or endangered species:

The Bossou Forest Reserve is threatened by the decline in chimpanzee populations. The forest has been fragmented, partly from inappropriate farming practices and illegal logging of commercial trees for fuel and processing purposes, especially wildfires. There are farm expansion threats that encroach on conservation areas. This incentive is the clear dominance of agriculture as the primary source of livelihood for the communities. The very design of the Reserve where it is not clearly delineated from community settlements presents a continuous clear and present danger to long-term ecological management. Potential exploratory mining activities in the Nimba area are presenting incipient threats from future uncontrolled human habitation, watershed, and ecological destruction.

Availability of valuable resources (historic, recreational, educational, cultural, ecological, conservation, scientific) covering both latent and established areas:

The Bossou Reserve contains native fauna that are essential components of the Reserve's biodiversity. The Green Corridor Project is perceived as a vital attraction for recreational tourists and academics. The interaction between the Bossou and the adjoining communities promotes economic activities for the people and provides an increasingly diverse income for traders and conservation experts alike. In addition, communities such as Serengbara serve as access towns to the Mount Nimba Strict Nature Reserve. The natural bridge in Serengbara and Nyon could also serve as a tourist attraction if developed well. The rivers that drain the communities are considered critical to their domestic and agricultural activities. The communities believe that rivers are important and must be protected at all times. The plantations are a vital source of sustenance for the communities and a sustainable strategy for conserving the chimpanzee corridor against wildfires. It provides their food and nutritional needs and presents an opportunity to safeguard further degradation of the protected areas from annual farming practices.

Indigenous community views on BFR management:

The communities recognize the work of the existing management consortium managing the BFR and the Green Corridor Project. However, concerns were expressed about the lack of community participation and leadership in the management process, outside the tour guide services provided by the local people. Parallel interventions by the USFS and AUDER are welcome and present alternative approaches to building sustainable ecotourism and conservation development. To date, the USFS supported a Fire Management Brigade, and the recently formed women-only Fire Management Team reflects increasing community agency.

## 5. Discussion

A community-based approach to ecotourism planning recognizes both conservation priorities and the well-being and resilience of society. The contributory and participatory nature of the mixed methods we employed, including the mapping outputs, resulted in a vastly improved understanding of the human-forest interactions and conservation opportunities in the landscape that can translate into policy and management.

This paper reports on an effort to examine community-centered sustainable ecotourism planning in the BFR. It highlights the failure of previous and existing approaches to activate community agency and support in the management of the protected area. The methodology is heavily influenced by the LAC framework and complementary models, including a situational analysis and resource opportunities assessment. Although the study employed a modified application of the LAC framework, it has demonstrated the need for local voices in the development of the BFR. The concept mapping and PGIS confirm that the most obvious threat to the survival of the chimpanzees, according to community members, is the corridor—the large expanse of degraded lands separating the BFR from the wider Nimba Reserve. The greatest threat to the regeneration of the corridor is frequent wildfires, partly due to inappropriate farming, grazing, and hunting practices. There is evidence of enhanced community action to address the destruction of seedlings and young trees (in the corridor) and farmlands from wildfires. The USFS-supported Fire Management Brigade was successful in stopping the spread of two big wildfires in the first quarter of 2019. These among others have spurred the establishment of a spinoff Women's Fire Management Brigade, through the assistance of AUDER and IREB staff.

Mapping of baseline spatial data emerged from community conversation and interactions about locations and their importance. Sites were plotted using GPS units and their meanings and cultural and livelihood associations were documented. One strength of the outcomes of the PGIS is that it presents local-level correlations between biological and cultural data which might valorize both sets of values and the interconnections and tradeoffs between them in promoting conservation planning, sustainable ecotourism, and other livelihood sectors. We drew on the successes of other researchers employing PGIS in ways that combine qualitative

ethnographic data with quantitative GPS-located geographic data to enhance understanding of common pool resources and resource challenges [52,53]. The resultant maps were in our case shared with community leaders, returned and critiqued by community members who were the original contributors to produce an understanding of the shared livelihood resources, the changes in land use, and emergent conservation challenges.

Establishing community-centered participatory ecotourism planning in the Bossou Forest Reserve, the local communities already understand why it is critical to safeguard the forest ecosystem and other ecological resources as their livelihoods depend on these; pathways must be co-created to tap their contributions; there have to be knowledge-sharing and capacity-building opportunities to raise the awareness of all stakeholders; and local communities must be given the opportunity to communicate to the reserve management their ideas on social norms and safeguards around sustainable conservation and ecotourism practices. When incorporated into decision making, this geographic and spatial information could support priorities and decisions which reduce the vulnerability of this social-ecological system through locally informed conservation planning in this unique forested environment.

By applying concept mapping and PGIS to conservation and ecotourism planning in Bossou communities, we identified zones/areas to support livelihood activities and identified shortcomings with the conservation corridor designed to connect BFR with the greater Mount Nimba Reserve. We find that ecotourism activities with livelihood elements and land use planning would positively contribute to the effectiveness of the conservation corridor. The boundaries of the BFR are all potential areas where a change in land use could be implemented.

The positive outcomes of this study are rooted in the community members' own awareness of their role in shaping conservation outcomes. At the outset, existing geospatial data were insufficient. This study generated crucial maps that could serve as a basis for conservation and ecotourism planning using PGIS. Although doubt exists about the validity of citizen-generated spatial data, the maps developed through PGIS in this study had a reasonable accuracy in that they corroborated existing ecological data. This outcome can be attributed to residents' intimate knowledge of the landscape as well as their unique relationships with the chimpanzee population. In the mapping sessions, community members actively discussed the locations of points of interest and thoroughly reviewed the data visualizations of land uses in two-dimensional formats.

The on-the-ground maps that were produced complement the records of biodiversity and the context of ecosystem services. The maps also complement the conservation planning that has been established as part of the Mount Nimba Strict Reserve through the UNESCO guidelines for Biosphere Reserves. Under these guidelines, certain activities or land uses are prohibited. While certain areas are protected, wildfires, land erosion, and forest clearing in areas could be prevented through land use policies or changes in land use. It is widely accepted by stakeholders, including KUPRI, USFS, IREB, and the lead Governmental Agencies that reforestation with native trees and vegetation is needed to protect and support forest integrity and the functionality of the conservation corridor.

Pro-ecotourism responses were captured in this study based on a consensus among community members. Any ecotourism plan is considered voluntary. The green corridor is located outside of the Mount Nimba Strict Conservation Area, so the management has no jurisdiction over the BFR. Supporting the green corridor would result in an opportunity for communities via the increase in potential income from visitors interested in seeing a viable chimpanzee population. Land use decision making achieved through consensus is critically important to communities and a conservation plan that supports the success of livelihoods through ecotourism is preferable. The ecosystem plan can be instrumental.

Government entities, which actively promote forest and biodiversity conservation in Bossou Forest Reserve, should adopt leadership roles to support communities to develop alternative livelihood and ecotourism sectors [54]. The ecotourism plan has combined community consensus and conservation scientists' input and therefore could be instrumental in

resolving compatible livelihood activities. Ecotourism and recreation activities developed in the area should be undertaken with income generation and sustainability in mind. Based on field observations and experiences in the study area, the current ecotourism activities in Bossou are clearly contributing to community interest and involvement in the reserve, improving the stressors on local communities and the chimpanzee population.

The location of the communities near/in the conservation corridor of BFR is strategic for ecotourism activities because these communities are connected to the main road leading to the Mount Nimba Strict Forest Reserve, which is a UNESCO Natural World Heritage Site. Community members identified activities and subsidiary attractions in various areas surrounding the BFR. The Guinean Ministry of Tourism should formulate a long-term tourism plan for the greater area of BFR and Mount Nimba area.

While the initial success of this participatory ecotourism planning effort is recognized, several concerns raised by community members need to be addressed. The environmental planning efforts to enhance ecological connectivity need to be carefully considered in relation to land use. Community involvement is paramount, with full agreement on proposed zoning. Along these lines, we have three general recommendations from our research that could assist in the efforts of participatory planning and management of ecotourism in this area.

First, it is observed that the communities have adjusted well to the perceived growth of tourism in the BFR and the associated opportunities for revenue generation through ecotourism. Going forward, it is critical that the ecotourism potential of the place is explored fully and sustainably, within the limits of community acceptance. However, these will be accompanied by certain tradeoffs, such as restrictions to farming practices in a manner that might significantly alter historical and current lifeways yet sustain socio-economic gains for the communities.

Second, the study underscored the inherent difficulties in the corridor restoration process. This restoration process has not fully explored the participation of the local communities. To complement the growing community interest, we recommend the development of cash crop plantations around the Green Corridor Project—on the approaches to neighboring communities. These should be owned and managed by individual community members with their existing farms sharing boundaries with the green corridor. It is anticipated that this strategy will provide extra incentive to mitigate any wildlife destruction to the conservation corridor and complement solutions to the wildfire hazard through the protection of the plantations. Plantations could be a possible target for chimpanzees. However, it is anticipated that by the time the cash crops will mature for consumption, more naturally regenerated, and the secondary restored species will also start fruiting to serve as food for the chimps.

Third, there clearly appears to be a lack of proper cooperation between the three main supervising government entities: Ministry of Tourism, Hotels, and Handicrafts; Ministry of Environment; and Ministry of Higher Education and Scientific Research, which is the supervising ministry for IREB which manages the BFR. First-hand community interactions with policymakers and implementing agencies such as IREB point to gaps in policy coordination and coherence in the conservation and tourism sectors. It is important that the inter-sectorial linkages are strengthened to materialize and maximize their potential benefits. Guinea exhibits a deep deficit of capable governance institutions in the context of conservation and ecotourism, a deficit that outside groups are unlikely to fill and that will torpedo all the well-intentioned efforts and community-based planning this study supports. Institutional weaknesses are reflected in the management of BFR and the adjoining ecotourism area.

## 6. Conclusions

In the Bossou Forest Reserve, the unplanned land use and livelihood activities of local communities have increasingly strained the biodiversity and ecosystem connectivity of this mountainous area. Owing to the absence of spatially explicit information, collecting

land use-related data and insights from local communities has proven to be fundamental to locally informed and conservation-oriented ecotourism planning [55]. The positive outcomes of this study were linked to the meaningful engagement of diverse stakeholder groups in informing future ecotourism planning. The ecotourism planning process emphasized spatially explicit livelihood and conservation concerns with a participatory approach that included the involvement of residents in concept mapping and PGIS. By connecting locations and specific sites to livelihood and land use concerns as well as biocultural values, the PGIS sought to improve the understanding of the relationships between these critical dimensions of the social-ecological system specific to the Bossou landscape. This was the first participatory ecotourism planning process of its kind in the area of the Mount Nimba Strict Forest Reserve. The spatial data generated through this study will serve as a baseline for monitoring future land use and livelihood activities through the analysis of multi-temporal satellite images. Community-based ecotourism planning with concept mapping and PGIS could be applied to support conservation and sustainable planning objectives in other socio-ecological settings.

**Author Contributions:** D.S.: Field research, data collection, analysis, preparation of original draft; K.B.: original funding acquisition, project administration, conceptualization, field facilitation, writing—review and editing; S.J.H.: study design, conceptual framing, analysis, writing—review and editing. All authors have read and agreed to the published version of the manuscript.

**Funding:** Research was funded through the United States Forest Service through a grant to the University of Montana (14-DG-11132762-409 grant number 364996). Contributions to support the authors' time and effort were provided by the University of Montana. An ongoing series of projects to support Bossou's agricultural communities and ecotourism planning was also supported by USAID.

**Institutional Review Board Statement:** Ethical review and approval were waived for this study due to its "Exempt" status under the University of Montana Office of Research and Sponsored Programs.

**Informed Consent Statement:** Verbal informed consent was obtained from all subjects involved in the study.

**Data Availability Statement:** The resource maps and data that are referred to are available on request from the corresponding author.

**Acknowledgments:** We would like to thank the Guinean communities for their interest and for taking the time to share their insights and values that they associate with this forest landscape. We would like to extend our many thanks to Kevin McManigal for technical guidance and mentorship, Hannah Shafer for cartographic contributions, and Sydney Qualls, the lead technician who systematized geodata and converted the participatory maps into digital maps for use in GIS. We are grateful to Ali Gaspard Soumah and Sophie Demartini for their field assistance and contributions to data collection and community engagement and to Steven Levine for his editorial expertise. We appreciate the various funding organizations that have made this work possible and the many community members who are dedicated to conservation in the Nimba Mountains.

**Conflicts of Interest:** One of the authors is the West Africa Regional Advisor, USFS. The funders of the research had no role in the design of the study; in the collection, analysis, or interpretation of data, in the writing of the manuscript, or in the decision to publish the results.

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
