# Peer review of "Community-Centered Sustainable Ecotourism Planning in the Bossou Forest Reserve, Guinea, West Africa"

_sustainability, doi:10.3390/su15054615_

Round 1

Reviewer 1 Report

Dear Editor and authors,

This outstanding study was created using extensive field research and a literature review. Such conservative studies with both the government and locals should be promoted in all fields for the conservation of globally threatened species.

This study is of the utmost importance, especially in ecotourism planning, with the participation of local communities in the conservation of Bossou chimpanzees. It provides an excellent review with complete knowledge of the conservation of the Bossou chimpanzees. It will also allow comparative considerations and conclusions in the future, especially since climatic changes, local participation, and government efforts to conserve the Bossou chimpanzee populations.

It perfectly meets the objectives of Sustainability; I, therefore, recommend its publication.

As a reviewer, I have no comment on this perfectly written and well-prepared manuscript. They meet the current international standards.

Reviewer 2 Report

The present study offers an interesting and valuable insight into one of the key attributes of the issue of Community-based ecotourism planning with concept mapping and PGIS, that could be applied to support conservation and sustainable planning objectives in other socio-ecological settings. The study is beneficial in the field of planning and setting marketing e-communication strategies for hotels. However, I would like to make few remarks:  Theoretical conceptual frameworks are absent in the study. According to the main aim of the study, it is necessary to define (and refer to examples of research carried out in similar areas) the essence and principles of functioning of community-based ecotourism, the concept of poverty alleviation by developing tourism in local communities, from the perspective of framing the design of strategic frameworks for the development of tourism in local poor communities in Guinea. It is necessary to give a brief outline of the principles of functioning of participatory place management based on Participation of local communities (see, for example, the study of Destination Management Based on Participation of Marginalized Communities: Matlovicova, K. et al. 2016. Selected Theoretical Aspects of the Destination Marketing Based on Participation of Marginalized Communities. 8th International Annual Scientific Conference on Hotel Services, Tourism and Education, pp. 128-143) as well as the frameworks of the concept of Community-Based Ecotourism (CBET - see Demkova, M. et. al. Potential for Sustainable Development of Rural Communities by Community-Based Ecotourism. A Case Study of Rural Village Pastanga, Sikkim Himalaya, India. Geojournal of Tourism and Geosites, 43/3, 964-975; and see also: Matlovicova, K. et al. 2022. Stimulating Poverty Alleviation by Developing Tourism in Marginalised Roma Communities: A Case Study of the Central Spiš Region (Slovakia). Land, 11, 1689). Furthermore, it is necessary to at least briefly outline the basic principles and different policy approaches of the local governments, and NGOs to tourism development in Guinea. (see Klimovsky, D. et. al. Al. 2016. Roma settlements and poverty in Slovakia: Different policy approaches of the state, local governments, and NGOs. Anthropological Notebooks, 22/1, 23-42). Personally, however, I think this is a very interesting topical issue, based on a well-developed and original methodology for obtaining and processing relevant data. The paper has a logical structure, relies on relevant sources and provides an interesting perspective on the undoubtedly widely discussed problem of the Community-based sustainable ecotourism planning.

The above comments in no way diminish the quality of the study.

I recommend it for publication after minor changes.

Reviewer 3 Report

The effort of this work is evident and commendable, however, it requires more work in the structuring of the article. The figures must be improved, emphasizing the methods followed in the selection of sample size, determination of the accuracy of the maps.

Also try to avoid taking for granted, some things that were not developed by the study.

Although there is talk of natural resources, the study only focuses on the case of chimpanzees, it is not bad, but it would be convenient to say that as a first approximation.

From Table 1, how was the number of participants in each community reached and is this number representative?

In the results presented, some are not derived from the study but from a bibliographic review, this should go in background and put them as results of this study, for example lines 696 - 714.

How did this come about? 844 It is a statement that is based on what?

From the line 857-866 are results without discussion of the same

What precision tests were carried out to support what was said in lines 850

894 which subsidiary activities

I consider that the discussion of results is not a pertinent place to make recommendations.

What was said in lines 906 to 912 as stated

In the third suggestion, I do not know, although it may be a reality, no studies, etc., were made to support this statement.

954 as he maintains that it should be by satellite images and not by other processes such as continuous use of PGIS.

Round 2

Reviewer 3 Report

I had already sent a review accepting the article with the corrections given by the author(s). A file was attached where it was emphasized how interesting and important this type of approach was and that in the future it would be convenient for them to be supported with a more intense development of statistics in order to ensure representativeness of the surveys.